# Boosting AlphaFold protein tertiary structure prediction through MSA engineering and extensive model sampling and ranking in CASP16
Jian Liu[1,2], Pawan Neupane[1,2] & Jianlin Cheng [1,2] ✉

AlphaFold2 and AlphaFold3 have revolutionized protein structure prediction by enabling high-accuracy structure predictions for most single-chain proteins. However, obtaining high-quality predictions for difficult targets with shallow or noisy multiple sequence alignments and complicated multi-domain architectures remains challenging. We present MULTICOM4, an integrative structure prediction system that uses diverse MSA generation, large-scale model sampling, and an ensemble model quality assessment strategy to improve model generation and ranking of AlphaFold2 and AlphaFold3. In the 16th Critical Assessment of Techniques for Protein Structure Prediction, our predictors built on MULTICOM4 ranked among the top out of 120 predictors in tertiary structure prediction and outperformed a standard AlphaFold3 predictor. Our best predictor achieved an average TM-score of 0.902 for 84 CASP16 domains, with top-1 predictions reaching high accuracy (TM-score>0.9) for 73.8% and correct folds (TM-score>0.5) for 97.6% of domains. For best-of-top-5 predictions, all domains were correctly folded. The results show that MSA engineering using different sequence databases, alignment tools, and domain segmentation along with extensive model sampling is critical to generate accurate structural models. Combining complementary QA methods with model clustering further improves ranking reliability. These advances provide practical strategies for modeling difficult single-chain proteins in structural biology and drug discovery.

Since deep learning was applied to protein structure prediction in 2012[1], it has made substantial progress over the last 13 years[2–7]. In particular, in 2020, AlphaFold2[8] revolutionized the field by using a transformer architecture[9] to accurately predict tertiary structures for most single-chain proteins (monomers) in CASP14[10]. It was then extended to AlphaFold2-Multimer that can predict the quaternary structures of protein complexes more accurately than existing docking methods[11]. In 2024, AlphaFold3[12], based on a diffusion architecture[13], further improved the prediction of protein structure and interactions between proteins and other molecules.

Despite the general high accuracy of AlphaFold-based tertiary structure prediction, it still faces two significant challenges for some hard protein targets whose MSAs are shallow or noisy and do not contain sufficient inter-residue/inter-domain co-evolutionary information. The first challenge is to generate some high-quality structural models for them. The second challenge is to select good structural models after many models with different

folds are generated. AlphaFold self-predicted model quality scores (e.g., plDDT) cannot consistently select good/best models for hard targets. These challenges make it hard for users to obtain high-quality predictions for certain hard targets using AlphaFold.

To address these challenges, we developed MULTICOM4, an integrative protein structure prediction system that enhances AlphaFold2 and AlphaFold3-based tertiary structure prediction by improving its input and output. MULTICOM4 generates diverse MSAs using multiple protein sequence databases, different alignment tools, and domain-based alignments as input for AlphaFold to generate tertiary structural models. Moreover, it performs extensive model sampling to explore a large conformation space. Furthermore, it applies multiple, complementary model quality assessment (QA) methods (also called estimation of model accuracy (EMA) methods) and model clustering to rank and select final predicted structures.

[1]Department of Electrical Engineering & Computer Science, University of Missouri, Columbia, MO, USA. [2]NextGen Precision Health, University of Missouri, Columbia, MO, USA. ✉e-mail: chengji@missouri.edu

In the 2024 community-wide CASP16 experiment, our best performing predictor (name: MULTICOM) based on MULTICOM4 ranked 4th among 120 predictors in tertiary (monomer) structure prediction. If multiple predictors from the same group are counted once, it ranked 2nd among all the participating groups. It also consistently outperformed the standard AlphaFold3 server. The results demonstrate that integrating AlphaFold2 and AlphaFold3 structure predictions, diverse MSAs, extensive model sampling, and multiple model ranking methods can further improve tertiary structure prediction, particularly for hard targets.

Our results also show that, with diverse MSAs and extensive model sampling, AlphaFold 2 and 3 can generate models of correct folds for all CASP16 tertiary structure prediction targets. However, existing QA methods still cannot select correct models as top-1 prediction for all targets. Therefore, model ranking can be harder than model generation for hard targets. It is a major challenge that needs to be addressed in the future.

## Results

### Overall performance of the MULTICOM predictors in CASP16

During CASP16, 120 predictors participated in the protein tertiary structure prediction challenge. Predictors were required to predict the structure given the sequence of a protein target and send up to five predicted structures per target back to CASP, but the final evaluation is based on the domain(s) of the target. In total, 59 protein targets with 85 domains were released for prediction. In this evaluation, T1249v2 (T1249v2-D1) was excluded because its native structure is not available to us, resulting in 58 protein targets with 84 domains. It is worth noting that the 84 domains are considered evaluation units that should be evaluated as a whole according to the official CASP16 definition, which may not exactly correspond to the independent structural domains. Sometimes, two or more structural domains of a target are treated as one single evaluation unit because the orientation between them can be predicted well by some CASP16 predictors. The evaluation can be performed on either top-1 predictions or best-of-five predictions for a predictor. In this work, we mostly report the results of top-1 predictions. It is worth noting that this study focuses exclusively on tertiary structure prediction for monomeric targets in CASP16. The evaluation of interface-level accuracy of our CASP16 protein complex structure prediction can be found in another study[14].

To quantify the CASP16 predictors' relative performance, Z-scores were used to evaluate each predictor's top-1 prediction for each domain. Following the official CASP16 evaluation protocol, the original GDT-TS score of each model was converted into Z-score based on the average and standard deviation of the GDT-TS scores of all top-1 models of all the predictors. Then models with Z-scores lower than -2 were excluded to eliminate outliers. The Z-score of each remaining model was then recalculated based on the mean and standard deviation of the remaining models' GDT-TS scores. The performance of each predictor was determined by the sum of the Z-scores across all the domains. Following the official CASP16 evaluation protocol, only Z-scores greater than 0 were accumulated for each predictor. Moreover, to avoid redundant contribution from alternative conformations of the same domain, we averaged Z-scores for 18 alternative domain pairs (e.g., T1228v1 and T1228v2, T1239v1 and T1239v2, T1294v1 and T1294v2) to be included into the sum. The sum of the Z-score of 75 unique domains from all the 84 domains is used to compare the CASP16 predictors.

Figure 1A illustrates the accumulated Z-scores of top 20 out of 120 CASP16 predictors. Our MULTICOM predictor achieved a cumulative Z-score of 33.39, ranking 4th, right after three predictors (Yang-Server, Yang, Yang-Multimer) from the same group—the Yang group. If only the best predictor from each group is counted, MULTICOM ranked 2nd. The other MULTICOM predictors, including MULTICOM_LLM, MULTICOM_AI, and MULTICOM_human, ranked 10th, 17th, and 19th, with Z-scores of 30.98, 28.78, and 28.21, respectively. Notably, the standard AlphaFold3 predictor ran by Elofsson's group, AF3-server, ranked 29th with a cumulative Z-score of 25.71, indicating that MULTICOM predictors achieved substantial improvements over standard AlphaFold3. In this work,

we focus on analyzing the performance of our best-performing tertiary structure predictor - MULTICOM.

Figure 1B illustrates the Z-scores of MULTICOM on the 75 unique domains. A Z-score higher than 0.0 means that the top-1 model submitted by MULTICOM has better quality than the average quality of the top-1 models of all predictors. The average Z-scores for MULTICOM is 0.287. There are 4 domains with Z-scores greater than 1.0, 56 domains with Z-scores higher than 0 (above average quality), and 19 domains with Z-scores below 0 (below average quality).

Notably, MULTICOM achieved the highest Z-score of 3.81 for domain T1267s1-D1 (Fig. 2A), which is the first domain of the subunit T1267s1 of the complex target H1267 (stoichiometry: A2B2, a XauSPARDA protein). The structure of this domain was extracted from the complex structural model predicted for H1267 by the MULTICOM4 complex structure prediction module using AlphaFold3 with extensive model sampling. The high prediction quality for T1267s1-D1 may be because its highly stable and cohesive structural conformation was captured well by the extensive model sampling. In contrast, the Z-score for the second domain of the same subunit, T1267s1-D2, is –0.01, because it is involved in inter-chain interactions that were not correctly predicted in the protein complex model. An opposite example is T1237-D1 (Z-score = 1.23, Fig. 2B), a subunit of the tetrameric complex T1237o (stoichiometry: A4). The structure of this domain was extracted from the high-quality complex structure predicted by the AlphaFold3 predictor within the MULTICOM complex structure prediction module. These examples demonstrate that the accuracy of structural prediction for individual subunits in a complex target heavily depends on the quality of predicted complex structures.

For the monomeric target T1210-D1 (Z-score = 1.16, Fig. 2C), MULTICOM submitted a top-1 model generated by AlphaFold3 that achieved a GDT-TS of 0.724, the highest among all participating groups, outperforming the top-1 in-house AlphaFold2 model (GDT-TS = 0.636). The success on this target is due to the use of AlphaFold3 with extensive model sampling.

T1257-D1 (Z-score = 1.10, Fig. 2E), a subunit of the trimeric complex T1257o (stoichiometry: A3), presented a particularly challenging modeling case because T1257 is a long filament-like structure but AlphaFold3 tends to bend it to make the N- and C-termini interact. To overcome this problem, a divide-and-conquer strategy was applied to divide the complex structure into three overlapped regions, each of which was predicted independently using AlphaFold3. The models of the three regions were then combined to form a full-length straight complex model by superimposing them through the overlapped regions. It produced a high-quality top-1 model for both the complex and its subunit.

Even though MULTICOM performed above average on most domains, it underperformed on some domains with negative Z-scores, such as T1266-D1 (–0.09), T1271s7-D1 (–0.15), T1228-D1 (–0.28), T1235-D1 (–0.31), T1207-D1 (–0.32), T1271s3-D1 (–0.35), T1271s5-D2 (–0.35), T1239-D1 (–0.42), T1218-D2 (–0.55), T1226-D1 (–0.59), T1295-D1 (–0.60), T1271s8-D1 (–0.61), T1295-D2 (–0.92), T1239-D4 (–0.93), T1220s1-D1 (–0.97), T1218-D3 (–1.34) and T1218-D1 (–2.36) due to various reasons.

For T1266-D1 (Z-score = –0.09), a single-chain monomer target, the GDT-TS score of top-1 model of MULTICOM generated by AlphaFold3 0.797, which is below average. However, by using AlphaFold2 with a domain-based MSA, MULTICOM was able to generate a substantially better model. Specifically, the target was divided into two regions (residues 1–188 and 189–366) that were predicted as two separate domains. MSAs were generated for them separately and then concatenated as a full-length MSA for AlphaFold2 to generate a higher-accuracy model with GDT-TS 0.877. This example demonstrates that domain-based MSA generation can generate better MSAs for some targets to improve the prediction of the tertiary structure of AlphaFold, when the full-length MSA generation does not cover some regions of a target well. However, this high-quality model was not selected as no. 1 prediction by MULTICOM.

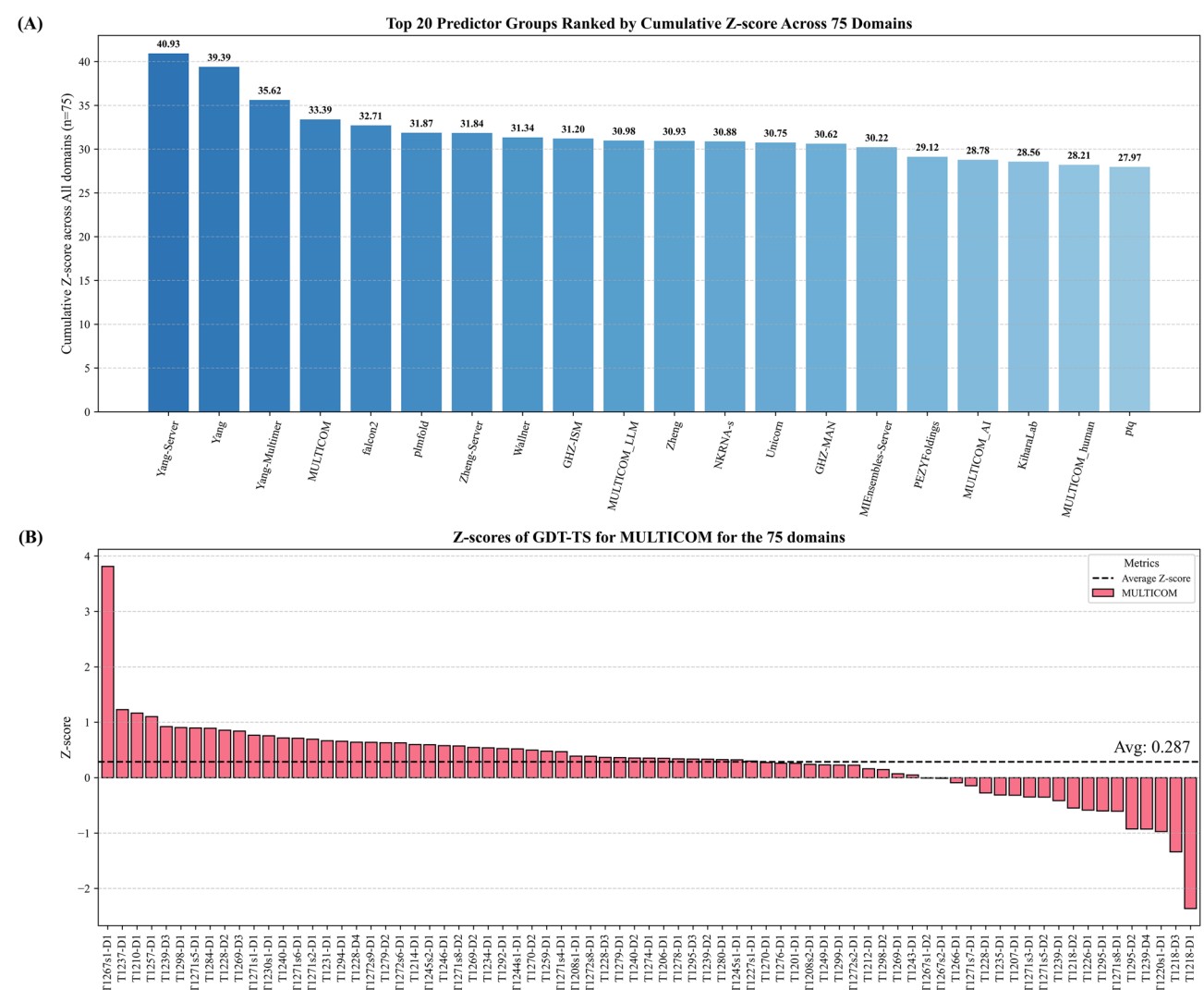

**Fig. 1 | The overall performance of top 20 out of 120 CASP16 predictors and the detailed performance of MULTICOM across 75 unique domains. A** The cumulative Z-scores of top 20 predictors. The standard AlphaFold3 predictor (AF3-server) run by Elofsson's group ranked 29th with a cumulative Z-score 25.71 (not shown); **B** The Z-score of the top-1 prediction of MULTICOM for each of the 75 domains.

For T1235-D1 (Z-score = −0.41), a subunit of T1235o (stoichiometry: A6), the top-1 model extracted from the AlphaFold3-predicted complex structure was of lower quality than one generated by AlphaFold2-Multimer. A similar issue occurred for domains T1295-D1 (Z-score = −0.60) and T1295-D2 (Z-score = −0.92), extracted from the octameric target T1295o (stoichiometry: A8), AlphaFold3-based models were of lower quality compared to those built using AlphaFold2-Multimer with diverse MSAs but were selected as top-1 prediction. Likewise, for T1220s1-D1 (Z-score = −0.97), a subunit of H1220 (stoichiometry: A1B4), the top-1 model extracted from the AlphaFold3-predicted complex structure with high predicted confidence (e.g., 0.594) had lower quality than the structure extracted from one AlphaFold2-Multimer prediction with much lower confidence (e.g., 0.384). These three examples signify the difficulty of ranking AlphaFold2 and AlphaFold3 models when they are not consistent.

For T1207-D1 (single-chain monomer target, Z-score = −0.32), it failed to predict helix-helix interactions within C-terminal region and its interaction with other regions. Instead, it predicted the C-terminal region as a long extended helix, resulting in a mediocre model (GDT-TS = 0.564, Fig. 2D). Although the full-length MSA that was used to build this structure contains 1310 sequences, the alignment depth for the last 30 residues of the C-terminus is much lower, ranging from 175 aligned sequences to as few as

2. The average depth across this region is only 84.4. The shallow coverage near the C-terminus likely caused the incorrect prediction for the region.

Different structural conformations were predicted for T1226-D1 (Z-score = −0.59) even though it has a deep MSA. But a heavily overrepresented yet incorrect conformation was selected as the top-1 prediction, while the correct conformation appeared only in a small number of models generated by AlphaFold3 was not selected. This example highlights the challenge of selecting good models when there are only a few of them in the model pool. The error of the top-1 model for T1226-D1 is similar to the top-1 model of T1207-D1, both of which incorrectly predicted C-terminal helical region as an extended helix rather than a folded helical structure.

For T1218-D1 (Z-score = −2.36), T1218-D2 (Z-score = −0.55), and T1218-D3 (Z-score = −1.34), the three domains of subunit T1218 in the complex T1218o, the quality of their structural models is relatively low due to the combination of the AlphaFold3-predicted complex structure and the complex model built from a significant complex template (PDB code: 4W8J) using Modeller[15]. The combination improved the quality of the complex model for T1218o but degraded the quality of the individual domains in the subunits of the complex. This example is consistent with the observation that AlphaFold-based prediction for individual domains is usually more accurate than template-based modeling.

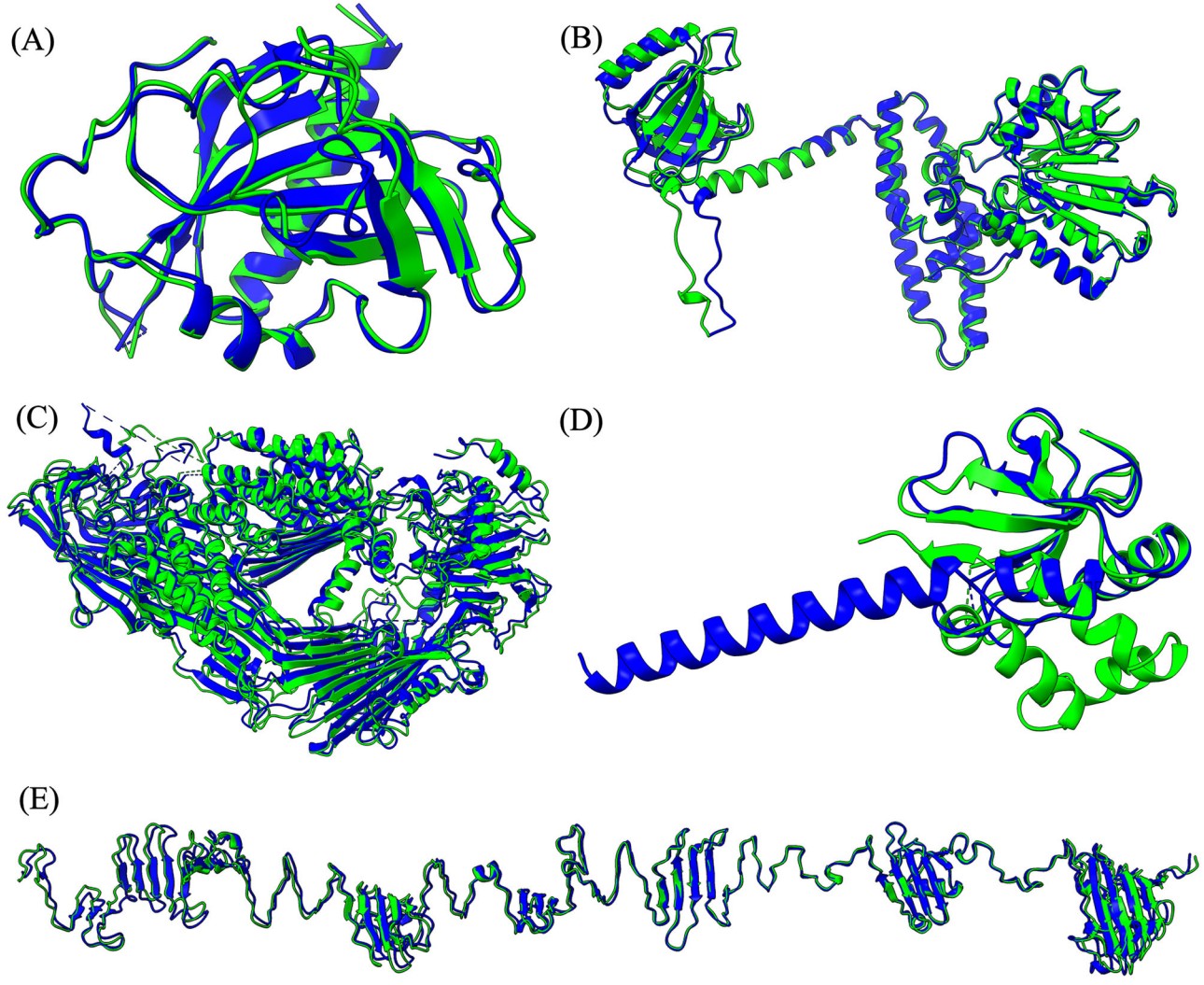

**Fig. 2 | The structural comparison between the native structures (green) and the top-1 models submitted by MULTICOM (blue). A** T1267s1-D1 (GDT-TS = 0.878), **B** T1237-D1 (GDT-TS = 0.905), **C** T1210-D1 (GDT-TS = 0.724), **D** T1207-D1 (GDT-TS = 0.564), **E** T1257-D1 (GDT-TS = 0.780).

Some domains in three protein-nucleic acids complexes including T1228-D1 (Z-score = −0.28) of M1228 (stoichiometry: A4B2C2D2E2), T1239-D1 (Z-score = −0.42), T1239-D4 (Z-score = −0.93) of M1239 (stoichiometry: A4B2C2D2E2) and T1271s3-D1 (Z-score = −0.35), T1271s5-D2 (Z-score = −0.35), T1271s7-D1 (Z-score = −0.15), and T1271s8-D1 (Z-score = −0.60) of M1271 (stoichiometry: A1B1C1D1E1F2G2H1I1J5R1) have low-quality top-1 predictions. The failure was largely due to the unsuccessful prediction of the regions of these domains in the complex structure models. Particularly, M1271 has more than 5000 residues, exceeding the length limit of AlphaFold3. It was divided into two sub-complexes whose structures were predicted separately and then combined. However, AlphaFold3 failed to predict some inter-chain interactions in the sub-complexes that influence the structures of the domains. Furthermore, some subunits such as T1271s5 and T1271s7 contain disordered residues, which may further complicate the prediction.

Figure 3 summarizes the TM-scores of top-1 models and best-of-top-5 models submitted by MULTICOM for all the 84 domains. The average TM-score was 0.902 for top-1 models and 0.922 for best-of-top-5 models. As shown in Fig. 3A, the top-1 predictions of MULTICOM achieved near-native quality (TM-score>0.9) for 62 domains (73.8%) and correct overall folds (TM-score>0.5) for 82 domains (97.6%). For only two domains, T1226-D1 and T1271s8-D1, the top-1 predictions had TM-scores below 0.5 due to the failure of model selection, while the best-of-top-5 predictions for them had TM-score above 0.5.

Figure 3B plots the TM-scores of top-1 models against the best-of-top-5 models submitted by MULTICOM. All the best-of-top-5 models for all 84 domains have a correct fold (TM-score >0.5). For most domains, the TM-scores of top-1 predictions are rather similar to those of best-of-top-5 predictions, indicating the model selection method worked well for them. However, for four domains (T1226-D1, T1271s8-D1, T1239v1-D3, and T1245s2-D1) highlighted in yellow, the score of the best-of-top-5-prediction is substantially higher than that of the top-1-prediction (the score difference >0.1), indicating MULTICOM failed to select high-quality models as top-1 models for these domains. In the case of T1226-D1, the top-1 model had a TM-score of 0.378, while an alternative and rare conformation present in a model with a TM-score of 0.725 was included in the top 5 models. The minority model generated by the extensive model sampling correctly folded the C-terminal helical region and its long-range interaction with other regions, while the majority models predicted the C-terminal region as a long, extended helix without interaction with other regions. For T1271s8-D1, a domain of a subunit of a very large complex target M1271, the best-of-top-5 model came from a partial complex model of M1271 predicted by AlphaFold3, which was different from the partial complex model containing the top-1 model. This example shows that when a complex model (e.g., M1271) is too big, dividing it into different sub-complexes to build models may help generate better structures for some subunits. Considering different options increases the chance of including good models into top 5 predictions. For

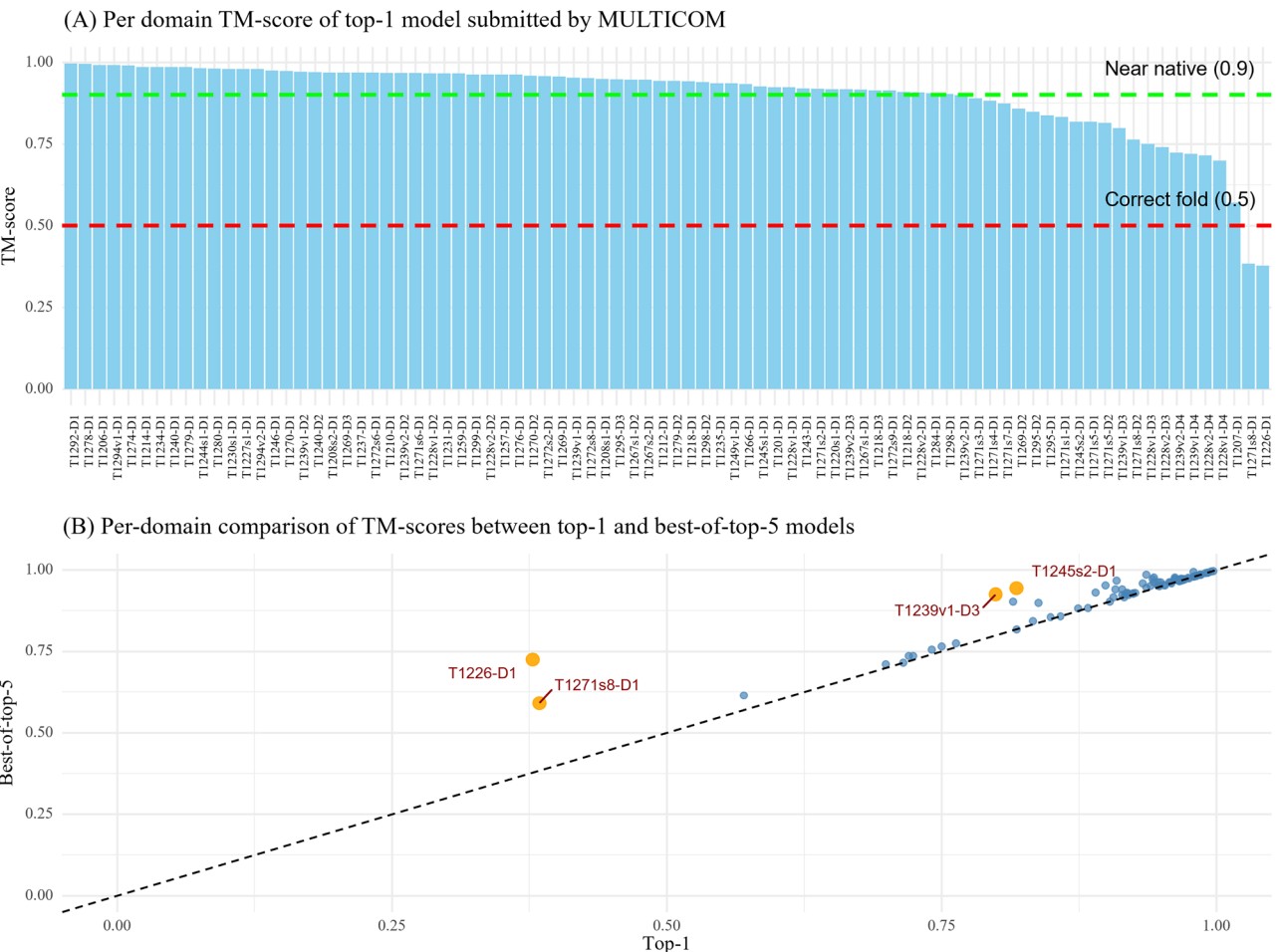

**Fig. 3 | The TM-scores of the top-1 and best-of-top-5 models submitted by MULTICOM for the 84 domains. A** The TM-scores of the top-1 models for the 84 domains. **B** TM-scores of top-1 models vs TM-scores of best-of-top-5 models on the 84 domains.

T1239v1-D3, the best-of-top-5 model and the top-1 model were selected from two different model clusters. For T1245s2-D1, a subunit of the H1245 complex (stoichiometry: A1B1), the top-1 model was predicted by AlphaFold3, which was much worse than the best-of-top-5 model predicted by AlphaFold2-Multimer using structure-alignment-based MSAs. The results demonstrate that ranking the best model as no. 1 for hard targets is still very challenging, but including alternative models into top 5 is an effective approach to increase the chance of obtaining at least a good model in the top five predictions.

**Performance comparison between MULTICOM and AF3-server**
Figure 4 compares the TM-scores of the top-1 and best-of-top-5 models of MULTICOM and the standard AlphaFold3 predictor (AF3-server) across 84 domains. Figure 4A plots the TM-scores of top-1 models of the two methods, while Fig. 4B plots the TM-scores of the best-of-top-5 models of the two methods. It is important to note that AF3-server generated exactly five models (e.g., one job) using AlphaFold3-server for most of the targets, which is much fewer than the number of models generated by MULTICOM.

In terms of top-1 model quality, MULTICOM achieved a slightly higher average TM-score (0.902) compared to AF3-server (0.891). Although this difference was not statistically significant according to a one-sided Wilcoxon signed-rank test ($p = 0.08$), MULTICOM exhibited stronger performance on 6 domains (i.e., T1271s8-D1, T1257-D1, T1267s1-D1, T1271s4-D1, T1239v2-D3 and T1284-D1), with a TM-score difference greater than 0.1, while AF3-server performed much better for 3 domains: T1271s5-D2, T1295-D2, and T1218-D2.

For T1257-D1, the superior performance by MULTICOM is attributed to the use of a divide-and-conquer strategy with AlphaFold3 that generated an elongated conformation, as opposed to the bent conformation predicted by default AlphaFold3, as described before. For T1267s1-D1, T1284-D1 and T1239v2-D3, the advantage of MULTICOM is due to the extensive sampling using AlphaFold3 and/or model clustering for improving model selection.

For the three domains derived from the same large complex M1271 (i.e., T1271s8-D1, T1271s4-D1, T1271s5-D2), MULTICOM performed better on T1271s8-D1 and T1271s4-D1, while AF3-server performed better on T1271s5-D2. The difference may partly stem from different subcomplexes used by the two methods to generate structural models.

In the case of T1218-D2, the worse performance of MULTICOM is due to its use of template-based modeling as described before. For T1295-D2, AF3-server performed better than MULTICOM, indicating that extensive model sampling does not alway lead to better top-1 models because generating more models does not mean good models can be ranked as no. 1.

In terms of best-of-top-5 models, the performance gap between MULTICOM and AF3-server widened (an average TM-score of 0.922 of MULTICOM vs 0.904 of AF3-server). The one-sided Wilcoxon signed-rank test confirms this difference as statistically significant ($p = 1.491\mathrm{e}{-05}$). MULTICOM produced substantially better models for 8 domains, while AF3-server performed much better on 2 domains. In addition to the 4 domains where MULTICOM already outperformed AF3-server in terms of top-1 model quality (i.e., T1257-D1, T1267s1-D1, T1239v2-D3, and T1284-D1), it performed much better than AF3-server on another four domains (T1226-D1, T1239v1-D3, T1245s2-D1, and T1295-D1 in terms of best-of-

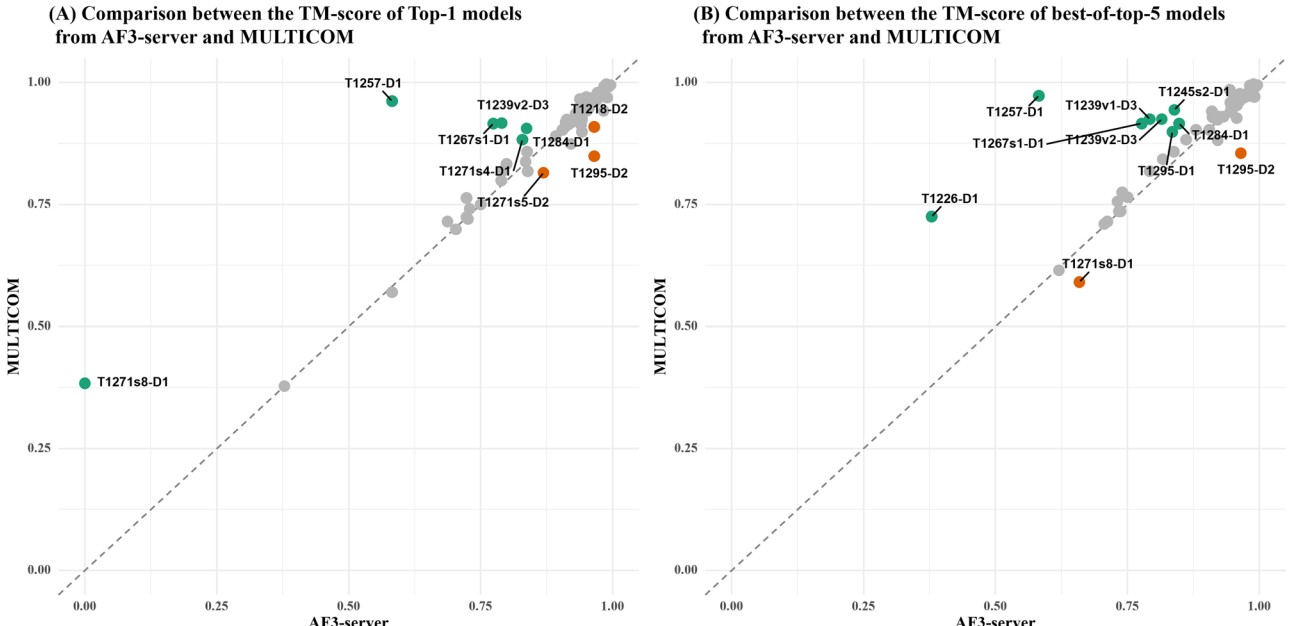

**Fig. 4 | Head-to-head TM-score comparison between MULTICOM and AF3-server on 84 protein domains. A** Top-1 models and **B** best-of-top-5 models.

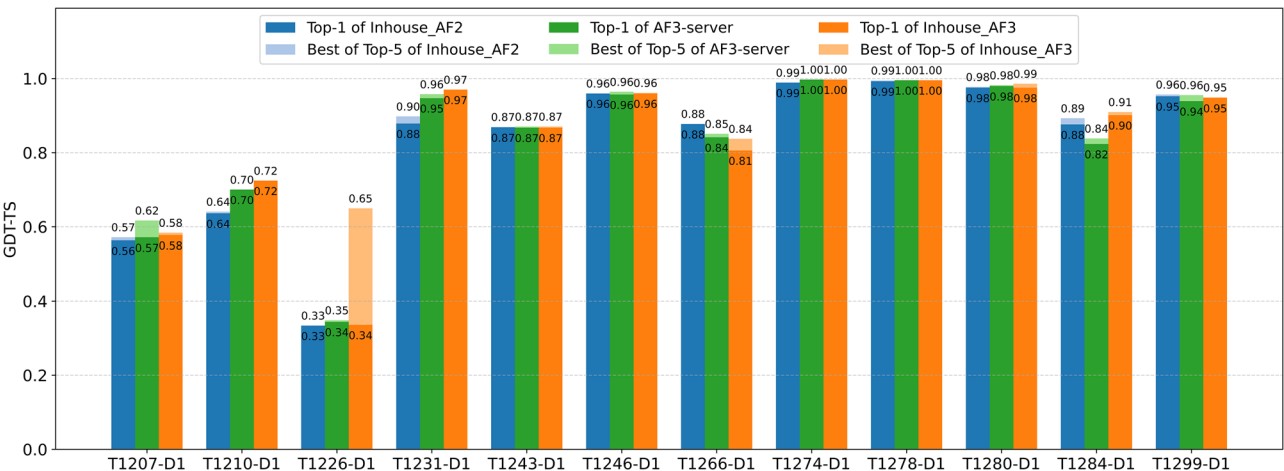

**Fig. 5 | Comparison of GDT-TS performance between the in-house AlphaFold2, the in-house AlphaFold3 predictor, and CASP16 AF3-server for the 12 single-chain monomer targets.** The plots show the top-1 and best-of-top-5 models generated by our in-house AlphaFold2, our in-house AlphaFold3 predictors, and AF3-server.

top-5 model quality. For T1226-D1, an alternative conformation that was included into the top 5 models had a significantly higher score than the models submitted by AF3-server. In the case of T1239v1-D3 (a multi-conformation target), the effective model clustering facilitated the selection of a higher-quality model. For T1245s2-D1, the structure extracted from the complex model predicted by AlphaFold2-Multimer that was included into top 5 models had better quality than that of AlphaFold3. A similar observation holds for T1295-D1, where the AlphaFold2-Multimer-based model outperformed AlphaFold3. However, for T1295-D2 and T1271s8-D1, AF3-server still outperformed MULTICOM, consistent with its advantage in top-1 model quality for those domains.

**Comparison of AlphaFold2 and AlphaFold3 on monomer targets**

To remove the impact of complex structure prediction on tertiary structure prediction, we compare our in-house AlphaFold2, our in-house AlphaFold3, and CASP16 AF3-server on 12 CASP16 monomer targets that are not a part of any multimer target. The GDT-TS scores of top-1 models and best-of-top-5 models of the three methods for the 12 domains are illustrated

in Fig. 5. In this comparison, the top 5 models for AlphaFold2 were selected according to its predicted plDDT scores, while AlphaFold3's ranking scores were used to select its top 5 models.

The average GDT-TS score of top-1 models for the in-house Alpha-Fold2 was 0.825, for AF3-server is 0.830, and for the in-house AlphaFold3 is 0.838. The average best-of-top-5 GDT-TS for the in-house AlphaFold2 is 0.830, for AF3-server is 0.840, and for the in-house AlphaFold3 is 0.870. The results show that the in-house AlphaFold3 performed better than AF3-server, while AF3-server slightly outperformed AlphaFold2. AF3-server sampled a much smaller number of structural models (e.g., five models) than the in-house AlphaFold2 (hundreds/thousands of models) but still outperformed it clearly demonstrates that AlphaFold3 performs better than AlphaFold2 in single-chain protein tertiary structure prediction.

In terms of top-1 predictions, the in-house AlphaFold3 performed similarly as the in-house AlphaFold2 for 9 out of 12 domains. For two domains (T1210-D1 and T1231-D1), the in-house AlphaFold3 substantially outperformed the in-house AlphaFold2 (GDT-TS score difference >0.05), while it performed substantially worse than the in-house AlphaFold2 on one

(A)  (B)  (C)  (D)

**Fig. 6 | Comparison of predicted structures for target T1226-D1. A** Native structure; **B** best model among the top-5 predictions of the in-house AlphaFold3 (GDT-TS = 0.650); **C** top-1 model of the in-house AlphaFold3 (GDT-TS = 0.336); **D** top-1 model of the in-house AlphaFold2 (GDT-TS = 0.334). The best-of-top-5 model of the in-house AlphaFold3 correctly predicted the folding of the C-terminal helices and their long-range interaction with other regions, while the other models did not.

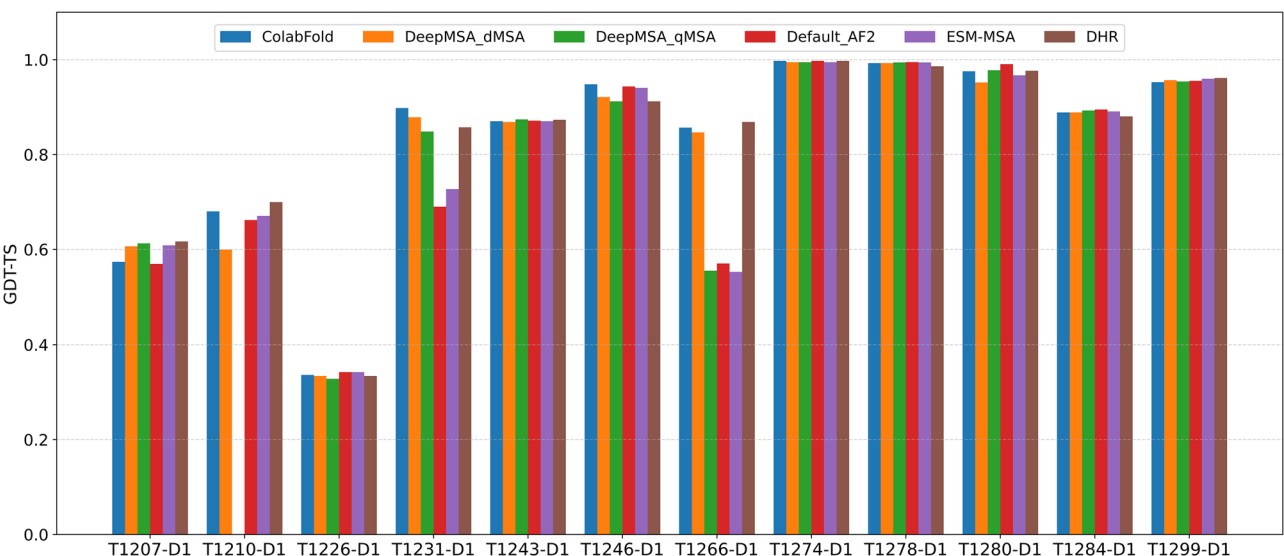

**Fig. 7 | Comparison of GDT-TS performance for top-1 models generated using different MSA sources.** Each MSA was evaluated using AlphaFold2 under identical input settings.

domain (T1266-D1). As described before, the in-house AlphaFold3's model for T1210-D1 is the best among all the CASP16 models submitted by all CASP16 predictors, indicating that the extensive model sampling with AlphaFold3 is essential to obtain high-quality models for this target. Notably, the in-house MSA for T1231-D1 was shallow and had less than 20 sequences, indicating that AlphaFold3 performed better than Alpha-Fold2 for some targets with limited evolutionary information. The in-house AlphaFold2 performed better on T1266-D1 because it used the improved domain-based MSA, while AlphaFold3 could only used the default MSA provided by the DeepMind's public AlphaFold3 web server.

In terms of best-of-top-5 models, there is only one major difference than the comparison in terms of top-1 models. For T1226-D1 (Fig. 6)—the hardest target for which the top-1 models of AlphaFold2 and AlphaFold3 had a low GDT-TS score between 0.33 and 0.35, but the best-of-top-5 in-house AlphaFold3 model (Fig. 6B) had a GDT-TS score of 0.65, sub-stantially higher than that of the top-1 model of the in-house AlphaFold3 (Fig. 6C) and the top-1 model of the in-house AlphaFold2 (Fig. 6D). This case highlights the advantage of AlphaFold3 over AlphaFold2 on certain challenging targets, where AlphaFold3 can generate some models with correct folds via extensive model sampling but AlphaFold2 fails. However, the best model was not ranked at the top partly because it was a minority model in the model pool. On this target, the best-of-top-5 model of the in-house AlphaFold3 also has a much higher GDT-TS score than that of AF3-server, indicating that extensive sampling is necessary for Alpha-Fold3 to generate good models for this hard target.

To further evaluate side-chain accuracy, we analyzed nine monomer targets for which top-rank models generated by each method (in-house

AlphaFold3, in-house AlphaFold2, and AF3-server) had high backbone quality (GDT-TS >0.70). Side-chain positioning was measured using the GDC_SC metric (Global Distance Calculation for side chains). Higher values indicate better accuracy. The results are shown in Supplementary Fig. S1. The in-house AlphaFold3 achieved the highest mean GDC_SC for both top-1 (0.644) and best-of-top-5 (0.647) predictions, followed by the AF3-server (0.631/0.638) and the in-house AlphaFold2 (0.618/0.620). Per-target comparisons show that AlphaFold3 substantially outperformed AlphaFold2 on T1231-D1 and T1274-D1 (GDC_SC improvements >0.07), while AlphaFold2 slightly outperformed AlphaFold3 on T1266-D1. Overall, AlphaFold3 can produce models with higher side-chain accuracy than AlphaFold2 for targets with high-quality backbone structure prediction.

### Performance of different MSAs for model generation

To investigate the impact of different multiple sequence alignments (MSAs) on structure prediction accuracy, we evaluated the accuracy of the structural models generated by the in-house AlphaFold2 using different MSA sources across the 12 single-chain monomer targets. All models were generated using identical structural templates and the same modeling parameters (i.e., num_recycle and num_ensemble) to isolate the effect of MSAs. The types of MSAs evaluated include ColabFold MSA provided by CASP16 and curated by Steinegger Group, DeepMSA_dMSA, DeepMSA_qMSA, Default_AF2, ESM-MSA, and DHR (see definitions of the MSAs in Section "Full-length MSA sampling") as shown in Fig. 7.

Each MSA above was used by AlphaFold2 to generate the same amount of structural models for the 12 single-chain monomer targets during CASP16. The top-1 model was selected by the AlphaFold2 plDDT

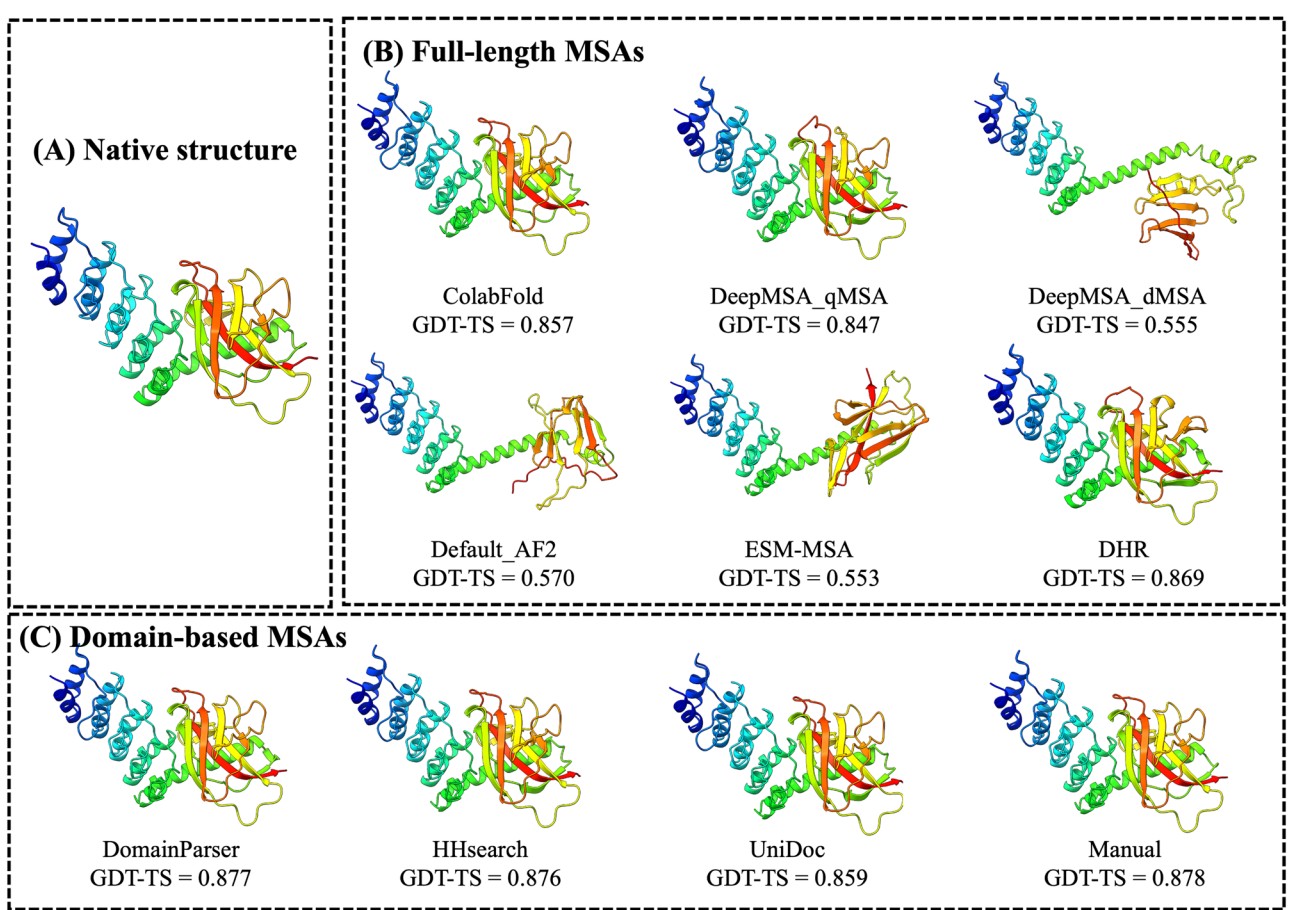

**Fig. 8 | Comparison of predicted structures for target T1266-D1 using full-length MSA and domain-based MSAs. A** Native structure; **B** Full-length MSAs, including ColabFold, DeepMSA_qMSA, DeepMSA_dMSA, Default_AF2, ESM-MSA and DHR; **C** Domain-based MSAs with different domain segmentation methods: DomainParser, HHsearch, UniDoc and Manual.

score. The average GDT-TS scores of these top-1 models for each MSA were as follows: ColabFold (0.831), DeepMSA_dMSA (0.820), Default_AF2 (0.790), ESM-MSA (0.793), and DHR (0.830). DeepMSA_qMSA were used to generate structural models for 11 targets during CASP16. Its average GDT-TS score over the 11 targets is 0.813. On average, ColabFold and DHR produced the most accurate models among the MSA methods tested. However, no MSA always performed better than other MSAs for all the targets, indicating it is useful to use them to generate a diverse set of models.

To complement backbone-level accuracy, we also evaluated side-chain positioning accuracy of the structural models generated from different MSAs using the GDC_SC metric (Supplementary Fig. S2). Across the seven targets with high backbone accuracy, the average scores were: ColabFold (0.657), DeepMSA_dMSA (0.644), DeepMSA_qMSA (0.647), Default_AF2 (0.657), ESM-MSA (0.647), and DHR (0.650). While the differences are small, ColabFold, Default_AF2, and DHR alignments tended to yield slightly higher side-chain accuracy for the targets.

To better illustrate the effect of MSA quality on prediction accuracy, we examined two representative cases: T1231-D1 and T1266-D1, where ColabFold and DHR MSAs substantially outperformed the default Alpha-Fold2 MSA (Default_AF2). For T1231-D1, all MSAs contained fewer than 20 sequences, yet the GDT-TS of the top-1 models varied significantly by different MSAs. The models generated using Default_AF2 and ESM-MSA performed noticeably worse than those using ColabFold, DeepM-SA_dMSA, DeepMSA_qMSA, and DHR, suggesting that these MSAs are of higher quality even though they were shallow. For T1266-D1, the impact of MSA source is also pronounced. Models from DeepMSA_qMSA, Default_AF2, and ESM-MSA consistently adopted the same low-quality

conformation, resulting in significantly lower GDT-TS scores compared to those from ColabFold, DeepMSA_dMSA, and DHR.

Interestingly, although T1266-D1 is formally categorized as one domain evaluation unit by CASP16 assessors and organizers, it actually has two structural domains (Fig. 8A) (a beta/alpha domain and a helical domain). Our domain-based MSA generation approach that generates MSAs for individual domains separately and then combines them with full-length MSAs also significantly improved model quality. As shown in Fig. 8B, the model predicted using the full-length default MSA (Default_AF2) has a GDT-TS of only 0.570. The main error occurs in the C-terminal domain and the inter-domain region, resulting in the partially misfolded C-terminal domain and incorrect domain orientation, possibly due to the lack of co-evolutionary signals in the MSA.

In contrast, the domain-specific MSAs generated by using domain segmentation tools such as DomainParser, HHsearch, UniDoc, and manual domain annotation (see Section "Domain-based MSA construction") yielded substantially better models than the full-length MSA, with GDT-TS scores ranging from 0.859 to 0.878 (Fig. 8C). These results indicate that for actual multi-domain targets, the domain-aware MSA sampling can improve the quality of MSAs and structural models and the quality of MSA is not sensitive to the specific domain segmentation tools used. This example demonstrates that MSA engineering is important for generating good predictions for hard multi-domain targets.

Finally, as shown in Fig. 7, all the MSAs failed on domain T1226-D1, even though they are deep. As discussed earlier, only AlphaFold3 with extensive model sampling was able to generate a model with correct fold for it. It shows that the extensive model sampling may be able to explore a large

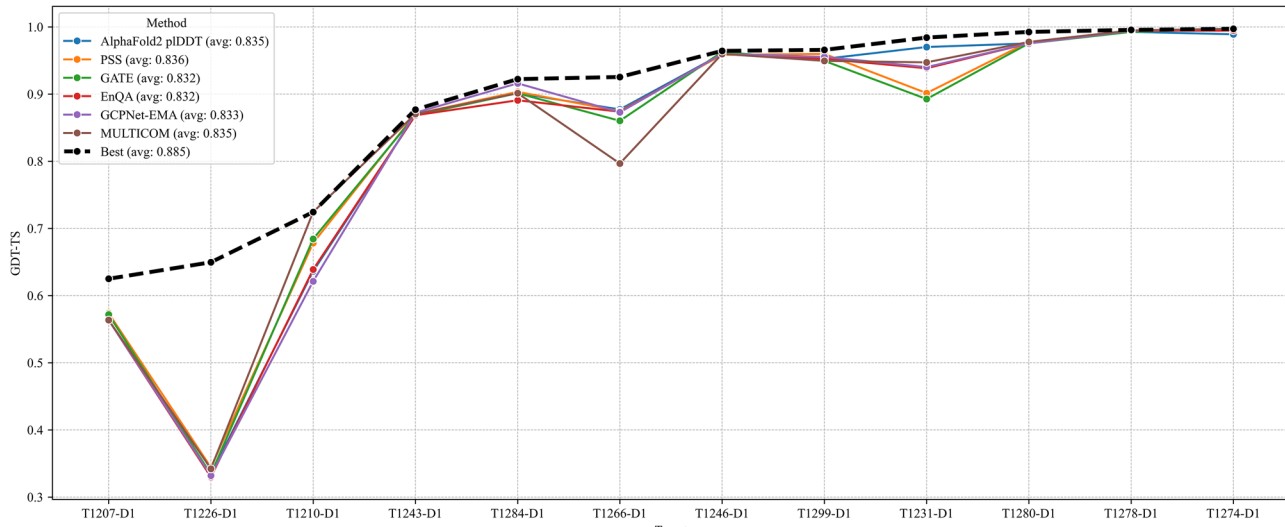

**Fig. 9 | GDT-TS scores of the top-1 models selected by different QA methods across 12 single-chain monomer targets.** The five QA methods include AlphaFold2 plDDT, pairwise similarity score (PSS), GATE, EnQA, and GCPNet-EMA. MULTICOM represents the result of the final top-1 models submitted by the MULTICOM predictor, which integrated the multiple QA methods. Best (the virtual line) represents the upper bound (highest) of the GDT-TS in the model pool for each target.

## Performance of different model quality assessment (QA) methods for model selection

We compare the performance of five individual QA methods in selecting top-1 models for 12 single-chain monomer targets, including AlphaFold2 plDDT, pairwise similarity score (PSS), GATE, EnQA, and GCPNet-EMA used in the MULTICOM system as well as MULTICOM itself.

The average GDT-TS scores of the top-1 models selected by the five individual QA methods are similar, i.e., PSS (0.836), AlphaFold2 plDDT (0.835), GCPNet-EMA (0.833), GATE (0.832), and EnQA (0.832). However, there are a significant difference between PSS and GATE ($p = 0.041$) as well as between PSS and EnQA ($p = 0.032$) according to the one-side Wilcoxon signed rank test. As shown in Fig. 9, the GDT-TS scores of the selected models have no or small difference for the five QA methods for 10 out of 12 targets, but vary a lot for two targets (T1210-D1 and T1231-D1). For T1210-D1, GDT-TS scores of the individual QA methods range from 0.621 (GCPNet-EMA) to 0.684 (GATE), while for T1231-D1, scores span from 0.893 (GATE) to 0.970 (AlphaFold2 plDDT). These differences highlight the variability in QA performance for some targets. Considering all the 12 targets, no individual QA method consistently performed better than the other QA methods.

We also examined the top-1 models submitted by MULTICOM. The average GDT-TS score across the 12 targets is 0.835, which is comparable to the best-performing individual QA methods. On T1210-D1, MULTICOM achieved the highest GDT-TS score of 0.724, outperforming all five QA methods, whose best score was 0.684 (GATE). For this target, the top-1 model was selected using an average of the GATE score and the AlphaFold3 ranking score, suggesting that combining complementary QA metrics can enhance model selection. However, this averaging strategy also has limitations. On T1266-D1, MULTICOM underperformed with a GDT-TS of 0.797, whereas all five QA methods achieved scores above 0.86. In this case, the averaging may have diminished the impact of stronger individual signals, leading to suboptimal selection.

It is worth noting that all the QA methods failed to select the good model for T1226-D1 because the best models belong to a small cluster in the model pool and all the QA methods selected models of mediocre quality from the dominant cluster as discussed before. This example suggests that clustering models into groups and comparing the characteristics of different clusters may help select models from less popular but correct clusters.

The findings above highlight that model selection is still a significant challenge and no single QA method can consistently select the best model across all targets. Each method performs well in certain scenarios but may struggle in others, depending on the characteristics of the target, such as MSA depth, structural complexity, or model diversity. Therefore, integrating multiple QA strategies, together with the information about model generation and model clustering, may offer a more robust solution for model selection.

## Discussion

The results of the MULTICOM4 system in CASP16 demonstrate that integrating extensive model sampling with AlphaFold2 and AlphaFold3, multiple MSA engineering strategies, and complementary QA methods improves tertiary structure prediction over the default AlphaFold3 or AlphaFold2. Notably, MULTICOM was able to predict the correct fold for 82 out of 84 domains (success rate: 97.6%) in terms of top-1 predicted models, and for all 84 domains (100%) in terms of the best-of-top-5 models, which is a significant improvement over the success rate of 90.43% of MULTICOM3 in CASP15[16]. The average quality of predicted structures for the 84 domains (average TM-score = 0.902) reached the accuracy of experimental structures.

Our experiment shows that AlphaFold3 outperformed AlphaFold2 in predicting tertiary structures of single-chain monomer targets on average. It is more suitable for generating accurate predictions for some targets with shallow MSAs and limited co-evolutionary information. For example, for two shallow-MSA targets T1231-D1 and T1284-D1 (MSA depth <20), AlphaFold3 generated models with higher GDT-TS than AlphaFold2. In the case of T1231-D1, the choice of MSA sources significantly impacted performance of AlphaFold2, but AlphaFold3 handled shallow MSAs better to generate accurate structures. These results are consistent with the design of AlphaFold3 that emphasizes MSA-feature generation less and the diffusion-based model generation more.

AlphaFold2's performance for shallow-MSA targets can be improved by MSA engineering such as domain-based MSA generation. For example, for T1266-D1, AlphaFold2 predictions using full-length MSAs yielded low-quality models (GDT-TS = 0.570), while predictions using domain-based MSAs achieved a high GDT-TS score of above 0.87. These improvements illustrate that AlphaFold2 is sensitive to the quality of shallow MSAs.

Another key insight from CASP16 is the importance, and also the limitation, of model ranking and selection. While AlphaFold is capable of generating rather accurate structures for all the CASP16 monomer targets, identifying the correct models from a pool of models remains challenging when most models in the pool are of low quality and similar. For instance, for T1226-D1, AlphaFold3 sampled both correct and incorrect conformations but assigned identical internal ranking scores to top ones in the two clusters. As a result, the incorrect structure from the predominant cluster was selected as the top-1 model. Although the correct model was present in the model pool, it was underrepresented and not prioritized by any of the quality assessment (QA) methods evaluated, including GATE, PSS, EnQA, GCPNet-EMA, and AlphaFold2 plDDT. This example illustrates a fundamental limitation of current QA strategies used in MULTICOM: when an incorrect conformation dominates the model pool, advanced QA tools may fail to identify the correct structure. Some manual inspection by human experts may help rescue the correct model before more effective QA methods are developed to tackle this challenge.

The challenge of selecting best/correct models as no. 1 prediction for hard targets necessitates including models with different conformations into top 5 predictions. In several cases, such as T1226-D1 and T1239v1-D3, high-quality models were not ranked first but were included in the top-5 models. This indicates that clustering models into groups and selecting one representative from each group is useful for selecting five models for hard targets.

Finally, the quality of the tertiary structure prediction for subunits from protein complexes largely depends on the quality of predicted complex structures. In comparison with protein complex (quaternary) structure prediction, the accuracy of tertiary structure prediction is generally higher. Ranking tertiary structural models is also easier than ranking complex structural models[17]. However, even though a predicted complex structural model is not accurate, the tertiary structure of individual units, particularly their domains, in the complex can usually be accurately predicted by AlphaFold2-Multimer and AlphaFold3.

## Methods
### Overview of the tertiary structure prediction module of MULTICOM4

MULTICOM4 is built on our previous version of protein structure prediction system (MULTICOM3)[16,18] and can predict both the tertiary structure of a single-chain protein (monomer) and the quaternary structure of a multi-chain protein complex (multimer). Given a protein target, if it is a standalone monomer, MULTICOM4 uses its tertiary structure prediction module to predict its structure. However, if it is a subunit (chain) of a protein complex (multimer), MULTICOM4 calls its quaternary structure prediction module to predict the structure of the complex first and then extract the tertiary structure of the subunit from the predicted complex structure. This approach can consider the interaction between subunits and generally produce more accurate tertiary structure predictions for subunits than predicting the tertiary structure of each subunit separately. MULTICOM4's quaternary structure prediction module has been reported in ref.[14]. Here, we focus on describing its tertiary structure prediction module.

The tertiary structure prediction module consists of several coordinated stages, as shown in Fig. 10: multiple sequence alignment (MSA) sampling, domain-based MSA construction for multi-domain proteins, structural template identification, deep learning-based model generation using three tools (e.g., AlphaFold2, AlphaFold3, ESMFold[7]), and model selection based on an ensemble of model quality assessment (QA) methods. The modular design of MULTICOM4 enables robust and scalable structure prediction across a wide range of sequence types and structural complexities.

### Full-length MSA sampling
The first step in the tertiary structure prediction module involves generating a diverse set of multiple sequence alignments (MSAs) to capture

evolutionary constraints and co-evolutionary signals. This MSA engineering process is critical for structure prediction.

The full target sequence is searched against several comprehensive protein sequence databases, including both general protein sequence databases (e.g., UniRef30_v2023_02[19], UniRef90_v2024_06[20]) and meta-genomic sequence databases (e.g., BFD[21,22], MGnify_v2023_02[23]), using both HHblits[24] and JackHMMER[25]. These searches yield MSAs with varying depths and taxonomic coverage, capturing diverse evolutionary signals across homologous sequences. In addition, the default MSA used by default AlphaFold2 (referred to as Default_AF2), which is generated from BFD, MGnify, and UniRef30, is also used.

Moreover, MULTICOM4 employs two additional tools: DeepMSA2[26] and Dense Homolog Retriever (DHR)[27] to generate more MSAs. DeepMSA2 constructs high-quality MSAs by iteratively searching a combination of general and metagenomic sequence databases, including TaraDB[28], Metaclust[22], MetaSourceDB[29], and JGIclust[30], using HHblits and JackHMMER. Three types of MSAs were generated by DeepMSA2 (i.e., DeepMSA_qMSA, DeepMSA_dMSA and DeepMSA_mMSA) using different search strategies, resulting in deeper and more diverse alignments that are especially useful for hard targets with few homologous sequences.

In parallel, MULTICOM uses DHR[27] that leverages embeddings generated by protein language models to search for homologous proteins to generate additional MSAs. DHR is particularly effective for enhancing alignment depth for targets with limited close homologs, helping to boost predictive accuracy for hard targets.

To enrich the MSAs further, we employed ESM2[7] to generate two synthetic sequences by systematically masking each residue in the input sequence, predicting replacements, and selecting the most confident substitutions. These synthetic sequences were appended to the Default_AF2 MSA, forming an extended alignment referred to as ESM-MSA, which improved the diversity and generalizability of sequence features used in structure prediction.

Finally, during CASP16, the Colabfold MSA provided by CASP16 and Steinegger group was used by MULTICOM4 to generate some structural models, some of which were selected as the sixth model submitted to CASP as requested.

### Domain-based MSA construction
For some multi-domain targets, full-length MSAs may cover some domains well but have low sequence depth for other domains because full-length sequence search may be dominated by some large domains or domains with many homologous sequences. To address this problem, for multi-domain targets, domain boundaries are identified using a set of domain prediction tools (DomainParser[31], UniDoc[32], HHsearch[33]). During CASP16, manual domain annotation was also tested. DomainParser and HHsearch rely on the protein sequence alone, whereas UniDoc and manual inspection uses an initial predicted protein structure to segment the protein into domains. To apply HHsearch for domain prediction for a target, the MSA of the target is used by HHsearch to build a profile represented as a hidden Markov model (HMM). The profile is searched against an in-house template profile (HMM) library built for the protein templates in the PDB. The search usually identifies some templates aligned with the target. The insignificant templates with e-value ( >1), sequence length (≤40 residues), or alignment coverage (≤0.5) are filtered out. Each continuous region of the target aligned with the remaining templates is treated as a separate domain, and any unaligned region longer than 40 residues is also considered a domain. In addition, DISOPRED3[34] is used to predict disordered regions, which is incorporated to refine and adjust domain boundaries for multi-domain targets.

For each predicted domain, independent MSAs were generated using the same procedure applied to full-length sequences, incorporating different MSA sources such as Default_AF2 and DeepMSA2_dMSA. Domain alignments in their MSAs were paired based on shared sequence identifiers. For the domain alignments that cannot be paired, gap padding was applied to each domain alignment to reconstruct a full-length alignment.

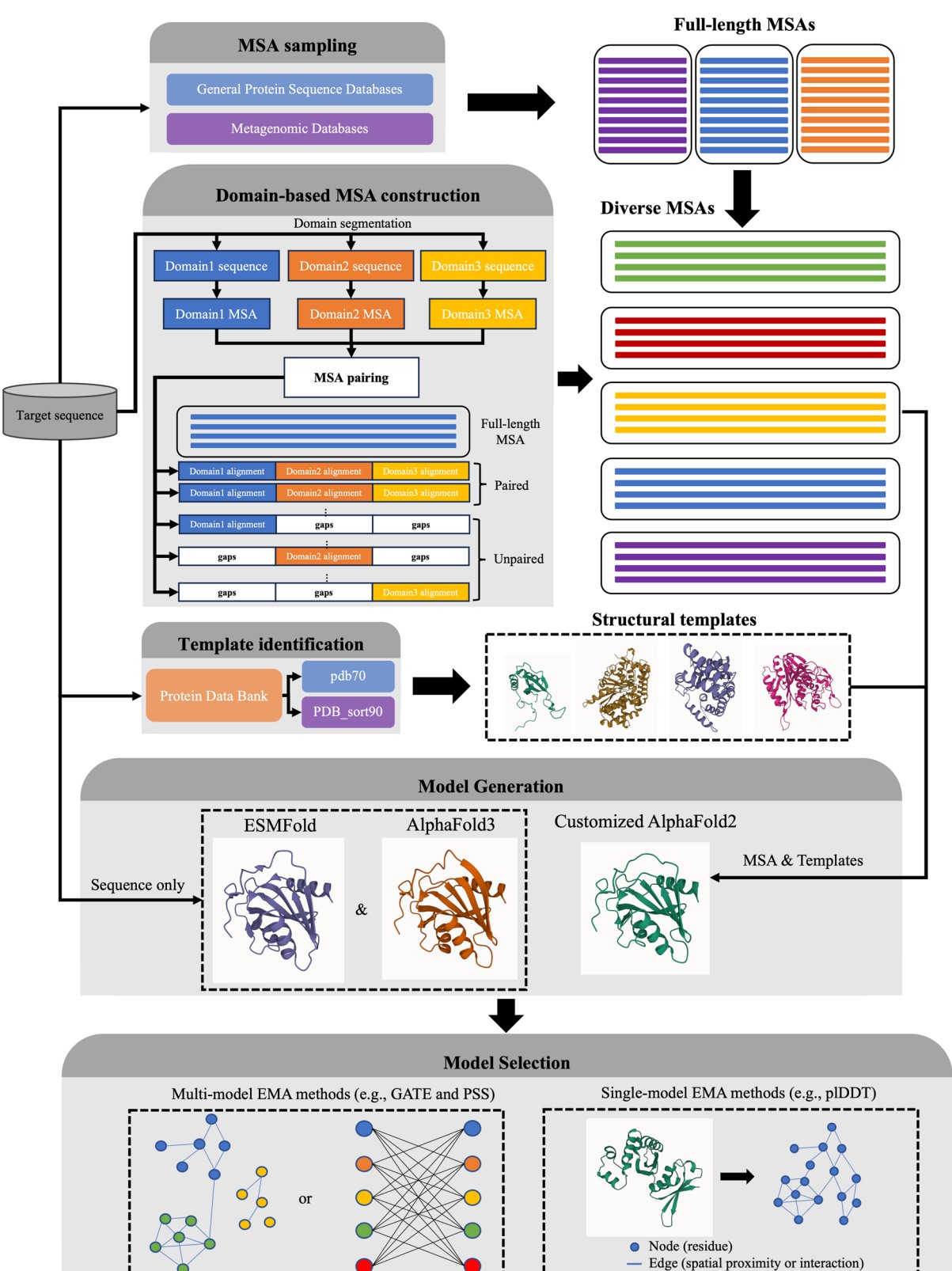

**Fig. 10 | Overview of the tertiary structure prediction module of MULTICOM4.** The module begins with multiple sequence alignment (MSA) sampling on both general protein sequence databases (e.g., UniRef30, UniRef90) and specialized metagenomic sources (e.g., BFD, MGnify) using different alignment tools. For multi-domain targets, in addition to generating full-length MSAs, domain-based MSA construction is performed by segmenting the target sequence into domains, generating individual domain MSAs, and combining them into full-length MSAs through alignment pairing and gap padding. Templates are identified by aligning sequence profiles against the PDB70 and PDB_sort90 databases. Structure prediction is carried out using multiple deep learning-based predictors, including a customized AlphaFold2 pipeline with various MSAs and templates as input (mainly), AlphaFold3 web server (extensive sampling, mainly), and ESMFold (for single-protein sequence modeling only). Predicted models are evaluated by multiple model quality assessment (QA) methods/metrics (e.g., GATE, PSS, and plDDT) to select high-confidence final predictions.

Each resulting domain-based MSA is combined with two full-length MSAs (i.e., Default_AF2 or DeepMSA2_dMSA), respectively, to form the final domain-based MSAs. Therefore, each domain-based MSA consists of three components: alignments from the full-length MSAs, paired domain alignments, and unpaired domain alignments with gap padding. This procedure was repeated across multiple combinations of domain prediction methods and full-length MSA sources, yielding 8 domain-based MSAs (four domain segmentation strategies × two MSA sources). These domain-based MSAs were used in parallel with the original full-length MSAs to improve alignment diversity and enhance the robustness of downstream structure prediction.

### Template identification
Templates are identified by aligning the sequence profiles of a target built from the MSA generated on UniRef90 against the template libraries curated from Protein Data Bank (PDB) using HHsearch. Two template libraries are employed: pdb70_v2024_03, a standard database included in AlphaFold2 that contains profile hidden Markov models (HMMs) for a representative subset of PDB chains filtered at a maximum of 70% sequence identity; and PDB_sort90[16], an in-house curated database filtered at 90% sequence identity to retain broader structural diversity. Utilizing both libraries increases the likelihood of identifying structurally informative templates, particularly in low-homology regions, and improves the quality of downstream folding predictions.

### Model generation
Structural models are generated using three deep learning-based predictors as follows. First, AlphaFold2 is run in parallel using different MSAs, with or without templates, and under varying configuration settings. These include changing the model preset (monomer or monomer_ptm) and enabling or disabling dropout in either the Evoformer or the structural module. The number of ensemble predictions and recycling steps are fixed at 1 and 8, respectively. For each target, ~2000 to 72,000 models were generated using AlphaFold2 during CASP16.

Second, to further explore conformational space, the DeepMind's AlphaFold3 web server was employed to generate hundreds to thousands of models during CASP16. As the software of AlphaFold3 has been released after CASP16 was concluded, it can be called by MULTICOM4 locally to generate structural models. For early-stage targets, such as T1207, where AlphaFold3 had not yet been fully incorporated into the pipeline, only 10 models were generated. For other targets, 200 to 5300 models were generated using AlphaFold3 during CASP16.

Finally, ESMFold is used to generate a small number of alternative models, enhancing structural diversity. Unlike AlphaFold2 and AlphaFold3, ESMFold only plays a very minor role in the MULTICOM4 prediction pipeline, with only 50 models generated for a few targets during CASP16. All models produced by AlphaFold2, AlphaFold3 and ESMFold are pooled together for model ranking and selection.

### Model selection
Model selection is based on a combination of AlphaFold's own confidence scores and independent quality assessment (QA) methods. Internally, each structure predictor (e.g., AlphaFold2, AlphaFold3, ESMFold) provides a global predicted lDDT score or its equivalent to estimate the confidence of each predicted structural model.

Because the self-estimated confidence scores cannot always select good structural models, MULTICOM4 applies several independent single-model or multi-model (consensus-based) quality assessment (QA) methods to rank predicted structural models. Two single-model QA methods, including EnQA and GCPNet-EMA, predict the quality of each individual model using deep learning approaches. In contrast, three multi-model QA methods, such as a graph transformer method—GATE, DeepRank3[33], and average pairwise similarity score (PSS)—consider the similarity between models in model quality assessment. The final model ranking can be generated by combining individual QA scores in different ways.

### The model ranking strategies of MULTICOM predictors in CASP16
During CASP16, MULTICOM4 participated in tertiary structure prediction as five predictors adopting different model ranking strategies. These included three server predictors: MULTICOM_AI, MULTICOM_GATE, and MULTICOM_LLM, as well as two human predictors: MULTICOM_human and MULTICOM. While all five predictors shared the same model generation pipeline, they differed in the strategies of selecting top five models for submission to CASP, which are described below.

- MULTICOM_AI: Models were ranked based on global plDDT scores. The top-ranked model was selected along with up to four additional models that were structurally diverse, defined by a TM-score less than 0.8 with each other. At least one model generated by AlphaFold3 was included if not already selected.
- MULTICOM_GATE: Models were ranked using GATE scores. K-means clustering was applied to group structurally similar models, and the top-ranked model from each cluster was selected. An AlphaFold3 model was included if it was not already represented in the final selection.
- MULTICOM_LLM: Initially, models were ranked based on the average of GATE and global plDDT scores. After AlphaFold3 models were incorporated into the system during the early stage of CASP16, the top-1 model was selected based on AlphaFold3 ranking score, while the remaining four models were chosen using the average of GATE and global plDDT scores.
- MULTICOM: This predictor initially used global plDDT scores for model selection. After AlphaFold3 was added into the system, the top-ranked model was selected based on the average of the AlphaFold3 ranking score and the GATE score. The remaining top models were selected using additional metrics, including the average of GATE and global plDDT scores, as well as GATE, GCPNet-EMA, EnQA, and PSS individually. Manual adjustment was applied when necessary to ensure diversity and quality.
- MULTICOM_human: Model selection initially relied on GATE scores. As the prediction pipeline evolved, multiple ranking metrics including global plDDT, GATE, GCPNet-EMA, EnQA, and PSS were considered. The top-1 model was selected using the average of GATE and global plDDT scores, with human oversight applied as needed. After AlphaFold3 was fully integrated, the model ranking primarily relied on global plDDT scores with manual refinement if required.

### Statistics and reproducibility
A total of 58 protein targets comprising 84 domains were used in the evaluation. Statistical comparisons between predictors were performed using the non-parametric one-sided Wilcoxon signed-rank test at a 95% confidence level.

### Reporting summary
Further information on research design is available in the Nature Portfolio Reporting Summary linked to this article.

### Data availability
The protein structures of CASP16 monomer targets are available at https://predictioncenter.org/download_area/CASP16/targets/. The protein structural models and analytical data generated in this study are available at https://zenodo.org/records/15588162[35]. Source data for the figures are provided in Supplementary Data 1. All other data are available from the corresponding author on reasonable request.

### Code availability
The source code of MULTICOM4 is available at: https://github.com/BioinfoMachineLearning/MULTICOM4[36].

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

## Acknowledgements
We thank CASP16 organizers and assessors for making the CASP16 data available. This work is partially supported by two NIH grants [R01GM093123 and R01GM146340].

## Author contributions
J.C. conceived the project. J.C., J.L. and P.N. designed the experiment. J.L., P.N. and J.C. performed the experiment and collected the data. J.L., J.C., and P.N. analyzed the data. J.L., J.C. and P.N. wrote the manuscript.

## Competing interests
The authors declare no competing interests.
