## [Transparent Peer Review file · Communications Biology]

Boosting AlphaFold Protein Tertiary Structure Prediction through MSA Engineering and Extensive Model Sampling and Ranking in CASP16

Corresponding Author: Professor Jianlin Cheng

Version 0:

Reviewer comments:

Reviewer #1

(Remarks to the Author)

Manuscript Title: "Boosting AlphaFold Protein Tertiary Structure Prediction through MSA Engineering and Extensive Model Sampling and Ranking in CASP16"

This manuscript presents the MULTICOM4 system, which integrates a range of strategies to enhance AlphaFold-based protein structure prediction accuracy. The system demonstrates strong performance in CASP16, a real-world blind test benchmark, making this work particularly noteworthy.

The most impressive aspect of the study is its comprehensive exploration of MSA (multiple sequence alignment) construction, combined use of AlphaFold2 and AlphaFold3, and large-scale model sampling coupled with multiple QA (quality assessment) techniques. By leveraging these strategies, the authors effectively overcome several limitations of existing AlphaFold models. The manuscript provides not only a detailed analysis of prediction accuracy across targets but also valuable insight into failure cases, offering practical guidance for researchers working in the field.

While the overall pipeline appears complex and may present a high entry barrier for users, the fact that the entire system is made publicly available on GitHub is a strong positive, as it supports reproducibility and community adoption.

Major Comments:

- Lack of side-chain accuracy evaluation:

The study focuses on backbone-level accuracy (GDT-TS, TM-score), but side-chain accuracy, which plays a crucial role in functional interactions, is not evaluated. A detailed analysis of side-chain positioning — particularly for high-quality models with GDT-TS > 70 — would add significant value to the study.

- No quantitative evaluation of interface accuracy in complexes:

The work would benefit from interface-level assessments using metrics such as DockQ score, especially for targets involving antibodies, nanobodies, or other biologically important interactions. Such an evaluation would enhance the completeness and impact of the manuscript.

Minor Comments:

- While the sampling strategies for complex prediction are well-described, it would be helpful if the authors could comment on the potential for interface-specific QA strategies.

- Can the authors discuss their views on MSA pairing in complex prediction? Additionally, when full-length MSAs are constructed from domain-specific MSAs, would integrating pairing strategies improve results? Any insight into this aspect would be appreciated.

Overall, this manuscript presents a well-organized and practical framework for improving protein structure prediction, offering

valuable methodologies for AlphaFold users and developers. The detailed MSA construction strategies and target-specific sampling techniques are particularly useful and transferable. For future work, incorporating side-chain accuracy assessment, interface-specific evaluation metrics, and antibody–antigen modeling considerations could further advance the precision and biological interpretability of structural predictions.

Reviewer #3

(Remarks to the Author)

Review comments for COMMSBIO-25-5390

This manuscript, “Boosting AlphaFold Protein Tertiary Structure Prediction through MSA Engineering and Extensive Model Sampling and Ranking in CASP16,” describes the performance of MULTICOM4 in the recent CASP16 structure prediction competition. MULTICOM4 achieved top-tier performance, ranking in the top 5 overall. As such, understanding the details of their methods and evaluating their predictions is of broad interest to the structural biology community, particularly given the current interest in structure prediction following the release of AlphaFold2. Specifically, the authors describe the details of their MSA generation process, which they maintain is the core explanation for their performance for certain hard targets. Understanding the myriad details and potential effects of groups attempting to build on these methods is crucial.

The manuscript assesses the performance method using the standard sum-of-Z-scores method used by CASP assessors. They display both the cumulative performance of the top 20 methods and their own performance across each of the 75 single-domain targets. They discuss the particulars of certain targets on which they performed well, as well as those on which they performed poorly. They note that the quality of predictions does seem to depend on the position of the domain in the quaternary structure of the experimental target, a not-uncommon occurrence in the post-CASP14 world. In general, the paper demonstrates that top-1 selection remains challenging even when a high-quality model is among the top 5 best models.

In addition to the CASP evaluation, the authors also did a comprehensive evaluation against AF3-server (useful to readers who may principally have access to publicly available tools) as well as an evaluation of multiple AlphaFold methods against CASP16 targets without significant inter-protein contacts. These data broadly support their claim that these methods generate similar models, and that the principal challenge is deriving a QA method that consistently supports the best model. The case study of T1266-D1 was particularly useful in driving home this point.

The methods description is comprehensive and goes into detail on the workflow for their construction of MSAs, which is a necessary part of any description AF2/AF3-based method, as this is the component most likely to vary. I saw no major issues in this section but I did see one minor issue:

1) For the domain-specific construction methods, DomainParser, UniDoc, and HHsearch are mentioned. DomainParser and UniDoc are well-documented domain partition methods, HHsearch itself is principally a protein-protein aligner. Its ability to be used as a domain partition method would depend on the library used. I'd like to see the specific search library described here.

Otherwise, the only other minor issues we saw were in the references:

2) The DeepRank3 reference incorrectly cites the Stenegger HHsearch paper. I'm not intimately familiar with the DeepRank development team, but I believe only DeepRank2 exists? Or may not be publicly available?

3) The ESM2 preprint [7] has since been published in Science: <https://doi.org/10.1126/science.ade2574>

Version 1:

Reviewer comments:

Reviewer #1

(Remarks to the Author)

I appreciate the authors' thorough responses and revisions.

- The inclusion of side-chain accuracy analysis using GDC_SC significantly strengthens the manuscript.
- The clarification of scope regarding interface accuracy—with reference to the authors' separate publication—is appropriate.
- The discussion of interface-specific QA and MSA pairing strategies is also thoughtful and forward-looking.

Overall, the revisions have improved the manuscript substantially, and I find it suitable for publication in its current form.

Response to Review Comments

Reviewer: 1

Major comments:

1. - Lack of side-chain accuracy evaluation: The study focuses on backbone-level accuracy (GDT-TS, TM-score), but side-chain accuracy, which plays a crucial role in functional interactions, is not evaluated. A detailed analysis of side-chain positioning — particularly for high-quality models with GDT-TS > 70 — would add significant value to the study.

Thank you for the great suggestion. We have conducted the side-chain accuracy evaluation using the GDC_SC (Global Distance Calculation for side chains) metric on the structural models of the single-chain monomer targets with high backbone quality (GDT-TS > 0.70). These results are now included in Sections 2.3 and 2.4 of the revised manuscript. All the changes are highlighted in red.

Our analysis shows that, AlphaFold3 generally produces models with higher side-chain accuracy than AlphaFold2, with the largest improvements observed for two targets T1231-D1 and T1274-D1. Among the different multiple sequence alignments (MSAs) tested with AlphaFold2, ColabFold, Default_AF2, and DHR alignments tended to yield slightly better side-chain positioning, although the differences were small.

2. - No quantitative evaluation of interface accuracy in complexes: The work would benefit from interface-level assessments using metrics such as DockQ score, especially for targets involving antibodies, nanobodies, or other biologically important interactions. Such an evaluation would enhance the completeness and impact of the manuscript.

Thank you for this insightful comment. We agree that interface-level metrics such as DockQ score are valuable for evaluating complex structures, particularly for biologically important interfaces. Because this manuscript focuses on tertiary structure prediction for monomeric targets, it only evaluates the accuracy of predicted tertiary structures of the CASP16 monomer targets using TM-score, GDT-TS score and newly added GDC_SC score, without performing interface-specific analysis for the quaternary structures of the complex targets.

However, we have evaluated the interface quality of the quaternary structures that our method (MULTICOM4) predicted for CASP16 complex targets using both DockQ and other interface metrics for protein complex prediction in another work. That work has been recently published in another paper (i.e., *Liu, J., Neupane, P., & Cheng, J. (2025). Improving AlphaFold2-and AlphaFold3-Based Protein Complex Structure Prediction With MULTICOM4 in CASP16. Proteins: Structure, Function, and Bioinformatics*).

To clarify this scope of this work, we add a sentence into the first paragraph in Section 2.1 to explicitly state that this work focuses on tertiary structure prediction and refer readers for the interface-level analyses to our CASP16 protein complex prediction article.

Minor comments:

1. - While the sampling strategies for complex prediction are well-described, it would be helpful if the authors could comment on the potential for interface-specific QA strategies.

We appreciate the suggestion and agree that the interface-specific QA is important for protein complex prediction. It helps distinguish models with similar overall global folds but with different inter-chain interface quality. Such interface-specific QA strategies can complement global quality measures such as TM-score as shown in our recently published complex prediction work (i.e., *Liu, J., Neupane, P., & Cheng, J. (2025). Improving AlphaFold2-and AlphaFold3-Based Protein Complex Structure Prediction With MULTICOM4 in CASP16. Proteins: Structure, Function, and Bioinformatics*).

The interface QA strategies are also relevant for tertiary structure prediction of monomer targets that are subunits of protein complexes. The interface QA strategies can help select better predicted quaternary structural models for protein complex targets, which usually automatically leads to selecting better tertiary structure predictions for the individual subunits of the complex targets.

2. - Can the authors discuss their views on MSA pairing in complex prediction? Additionally, when full-length MSAs are constructed from domain-specific MSAs, would integrating pairing strategies improve results? Any insight into this aspect would be appreciated.

Thank you for raising this important point. MSA pairing is an effective approach for capturing co-evolutionary signals between interacting chains in complex prediction. In the complex structure prediction module of our MULTICOM4 system, we have explored both enabling and disabling MSA pairing, and we have applied various pairing strategies, including the use of species information, UniProt accession IDs, STRING interaction data, and PDB protein complex records, to increase the diversity of input MSAs for AlphaFold-Multimer and thereby sample a wider range of complex structures. This usually improves the accuracy of protein complex prediction. However, for certain target types, such as antibody/antigen or nanobody/antigen complexes, where there is no co-evolution between subunits, we found it helpful to generate a large number of models with diverse parameters and settings, including using unpaired MSAs, to maximize conformational and interface diversity.

We also believe that integrating MSA pairing strategies with domain-specific MSAs could further increase structural diversity and improve the modeling of inter-chain or inter-domain orientations because many protein chains interact through their domains. It would be useful to consider both domain-specific MSAs, full-length MSAs, and inter-chain MSA pairing together to get further improved results. One possibility is to first concatenate domain-specific MSAs as full-length MSAs and then pair the concatenated full-length MSAs into the paired MSAs for protein complexes. Right now, there is no such method in the field to do this. We would like to explore this direction in the development of the next version of MULTICOM.

Reviewer: 2

The methods description is comprehensive and goes into detail on the workflow for their construction of MSAs, which is a necessary part of any description AF2/AF3-based method, as this is the component most likely to vary. I saw no major issues in this section but I did see one minor issue:

1) For the domain-specific construction methods, DomainParser, UniDoc, and HHsearch are mentioned. DomainParser and UniDoc are well-documented domain partition methods, HHsearch itself is principally a protein-protein aligner. It's ability to be used as a domain partition method would depend on the library used. I'd like to see the specific search library described here.

Thank you for the great question. Yes, using HHsearch for domain prediction depends on our inhouse template library. Specifically, to apply HHsearch for domain prediction for a target, the MSA of the target is used by HHsearch to build a profile (i.e., hidden Markov model (HMM)). The profile is searched against an inhouse template profile (HMM) library built for the protein templates in the Protein Data Bank (PDB). The search usually identifies some templates aligned with the target. The insignificant templates with e-value (>1), sequence length (≤ 40 residues), or alignment coverage (≤ 0.5) are filtered out. Each continuous region of the target aligned with the remaining significant templates is treated as a separate domain, and any unaligned region longer than 40 residues is also considered a domain. We have included this description in Section 4.3. All the changes are highlighted in red.

2) The DeepRank3 reference incorrectly cites the Stenegger HHsearch paper. I'm not intimately familiar with the DeepRank development team, but I believe only DeepRank2 exists? Or may not be publicly available?

The correct reference for DeepRank3 is “Liu, J., Wu, T., Guo, Z., Hou, J., & Cheng, J. (2022). *Improving protein tertiary structure prediction by deep learning and distance prediction in CASP14. Proteins: Structure, Function, and Bioinformatics*, 90(1), 58-72”. It was developed by us for the CASP14 experiment. The public code for DeepRank3 is available at: <https://github.com/jianlin-cheng/DeepRank3>. We have cited the correct reference in the revised manuscript.

3) The ESM2 preprint [7] has since been published in Science: <https://doi.org/10.1126/science.ade2574>

Thank you for noticing the problem. We have updated this reference in the revised manuscript.